# Linear Bandits with Non-i.i.d. Noise

## Abstract

We study the linear stochastic bandit problem, relaxing the standard i.i.d. assumption on the observation noise. As an alternative to this restrictive assumption, we allow the noise terms across rounds to be sub-Gaussian but interdependent, with dependencies that decay over time. To address this setting, we develop new confidence sequences using a recently introduced reduction scheme to sequential probability assignment, and use these to derive a bandit algorithm based on the principle of optimism in the face of uncertainty. We provide regret bounds for the resulting algorithm, expressed in terms of the decay rate of the strength of dependence between observations. Among other results, we show that our bounds recover the standard rates up to a factor of the mixing time for geometrically mixing observation noise.

## 1 Introduction

The linear bandit problem (Abe and Long, 1999; Auer, 2003) is an instance of a multi-armed bandit framework, where the expected reward is linear in the feature vector representing the chosen arm. More concretely, it is a sequential decision-making problem, where an agent each round picks an arm $X_t$, and receives a reward $Y_t = \langle \theta^\star, X_t \rangle + \varepsilon_t$, with $\theta^\star$ a fixed parameter unknown to the agent, and $\varepsilon_t$ zero-mean random noise. This framework has gained significant attention in the literature as it yields analytic tools that can be applied to several concrete applications, such as online advertising (Abe et al., 2003), recommendation systems (Li et al., 2010; Korkut and Li, 2021), and dynamic pricing (Cohen et al., 2020).

A popular strategy to tackle linear bandits leverages the principle of *optimism in the face of uncertainty*, via upper confidence bound (UCB) algorithms. The idea of optimism can be traced back to Lai and Robbins (1985), and its application to linear bandits was already advanced by Auer (2003). Since then, this approach has been improved and analysed by several works (Abbasi-Yadkori et al., 2011; Lattimore and Szepesvári, 2020; Flynn et al., 2023). This class of methods requires constructing an adaptive sequence of confidence sets that, with high probability, contain the true parameter $\theta^\star$. Each round, the agent selects the arm maximising the expected reward under the most optimistic parameter (in terms of reward) in the current confidence set. UCB-based algorithms have become popular as they are often easy to implement and come with tight worst-case regret guarantees.

For a UCB algorithm to perform well, it is necessary that the confidence sets are tight, which can be ensured by taking advantage of the structure of the problem. In this paper, our focus is on studying various assumptions on the observation noise. A commonly studied situation is when $(\varepsilon_t)_{t \geq 0}$ consists of a sequence of i.i.d. realisations of some bounded or sub-Gaussian random variable (see Lattimore and Szepesvári, 2020, Chapter 20). Often, the standard analysis can be extended to the case in which the realisation are not independent, but conditionally centred and sub-Gaussian (Abbasi-Yadkori et al., 2011). Yet, in real-world settings, this assumption is often unrealistic, as one can expect the presence of interdependencies among the noise at different rounds. For instance, in the context of advertisement selection, the noise models the ensemble of external factors that influence the

user's choice on whether to click or not an ad. The i.i.d. assumption implies that across different rounds these external factors are completely independent. In practice, the user choice will be affected by temporally correlated events, such as recent browsing history or exposure to similar content. Therefore, a more realistic assumption is to allow the dependencies to decay with time, rather than being completely absent. This way to model dependencies, often referred to as *mixing*, is common to study concentration for sums of non-i.i.d. random variables, with applications to machine learning (Bradley, 2005; Mohri and Rostamizadeh, 2008; Abélès et al., 2025).

In the present paper we relax the assumption that the noise is conditionally zero-mean in the bandit problem, and we allow for the presence of dependencies. Concretely, we replacethe standard conditionally sub-Gaussian setting with a more general formulation that accounts for conditional dependence of the noise on the past, by introducing a natural notion of *mixing sub-Gaussianity*. Within this context, we introduce a UCB algorithm for which we rigorously establish regret guarantees. There are two key challenges for our approach: constructing a valid confidence sequence under dependent noise, and deriving a regret upper bound for the UCB algorithm that we propose.

We derive the confidence sequence by adapting the *online-to-confidence-sets* technique to accommodate temporal dependencies in the noise. This approach, originally introduced by Abbasi-Yadkori et al. (2011) and recently extended and improved (Jun et al., 2017; Lee et al., 2024; Clerico et al., 2025), involves constructing an abstract online learning game whose regret guarantees can be turned into a confidence sequence. To deal with the dependencies in the noise, we modify the standard online-to-confidence-sets framework by introducing delays in the feedback received within the abstract online game. This approach is inspired by the recent work of Abélès et al. (2025) on extending online-to-PAC conversions to non-i.i.d. mixing data sets in the context of deriving generalisation bounds for statistical learning. There, a delayed-feedback trick similar to ours is employed to derive statistical guarantees (generalisation bounds) from an abstract online learning game.

For the regret analysis of the bandit algorithm, we also need to face some challenges due to the correlated observation noise. We address these by introducing delays into the decision-making policy as well. This makes our approach superficially similar to algorithms used in the rich literature on bandits with delayed feedback (see, e.g., Vernade et al., 2020a; Howson et al., 2023). These works consider delay as part of the problem statement and not part of the solution concept, and are thus orthogonal to our work. In particular, a simple adaptation of results from this literature would not suffice for dealing with dependent observations, which we tackle by developing new concentration inequalities. Another line of work that is conceptually related to ours is that of non-stationary bandits (Garivier and Moulines, 2008; Russac et al., 2019). In that setting, the parameter vector $\theta_t^\star$ evolves in time according to a nonstationary stochastic process, and the observation noise remains i.i.d., once again making for a rather different problem with its own challenges. Namely, the main obstacle to overcome is that comparing with the optimal sequence of actions becomes impossible unless strong assumptions are made about the sequence of parameter vectors. A typical trick to deal with these nonstationarities is to discard old observations (which may have been generated by a very different reward function), and use only recent rewards for decision-making. This is the polar opposite of our approach that is explicitly *disallowed* to use recent rewards, which clearly highlights how different these problems are. That said, there exists an intersection between the worlds of delayed and nontationary bandits (Vernade et al., 2020b), and thus we would not discard the possibility of eventually building a bridge between bandits with nonstationary reward functions and bandits with nonstationary observation noise. For simplicity, we focus on the second of these two components in this paper.

**Notation.** Throughout the paper, we will often use the following notations. For $u$ and $v$ in $\mathbb{R}^p$, we let $\langle u, v \rangle$ denote their dot product. $\|u\|_2 = \sqrt{\langle u, u \rangle}$ is the Euclidean norm, while for a non-negative definite $(p \times p)$-matrix $A$, $\|u\|_A = \sqrt{\langle u, Au \rangle}$ is a semi-norm (a norm if the matrix is strictly positive definite). For $r > 0$, $\mathcal{B}(r)$ denotes the closed centred Euclidean ball in $\mathbb{R}^p$ with radius $r$. Given a non-empty set $U \subseteq \mathbb{R}^p$, we let $\Delta_U$ denote the space of (Borel) probability measures on $\mathbb{R}^p$ whose support in $U$. Finally, $(u_t)_{t \geq t_0}$ denotes a sequence indexed on the integers, with $t_0$ its smallest index.

## 2 Preliminaries on linear bandits

We consider a version of the classic problem of regret minimisation in stochastic linear bandits, where an agent needs to make a sequence of decisions (or pick an *arm*) from a given contextual decision set

that may change over the sequence of rounds. We assume that the environment is oblivious to the actions of the agent, in the sense that the decision sets are determined in advance, and do not depend neither on the realisations of the noise nor on the agent's arm-selection strategy.

Concretely, we define the problem as follows. Let $\theta^\star \in \mathbb{R}^p$ be a parameter vector that is unknown to the learning agent. We assume as known an upper bound $B > 0$ on its euclidean norm (namely, $\theta^\star \in \mathcal{B}(B)$). Fix a sequence of decision sets $(\mathcal{X}_t)_{t \geq 1}$ in $\mathbb{R}^p$. We assume that for all $t$ we have $\mathcal{X}_t \subseteq \mathcal{B}(1)$. At each round $t$, the agent is required to pick an arm $X_t \in \mathcal{X}_t$, and receives the reward $Y_t = \langle \theta^\star, X_t \rangle + \varepsilon_t$. The sequence $(\varepsilon_t)_{t \geq 1}$ represents the random feedback noise. The noise across different rounds is typically assumed to be conditionally centred and to have well behaved tails. For instance, a common assumption is to ask that $\mathbb{E}[\varepsilon_t | \mathcal{F}_{t-1}]$ is centred and sub-Gaussian, where $\mathcal{F}_t = \sigma(\varepsilon_1, \ldots, \varepsilon_t)$ is the $\sigma$-field generated by the noise.[1] This is the assumption this work relaxes.

The agent aims to find a good strategy to pick arms $X_t$ that lead to a high expected $T$-round reward $\sum_{t=1}^{T} \langle X_t, \theta^\star \rangle$. To compare their performance to that of an agent playing each round the best available arm (in expectation), we define the *regret* after $T$ rounds as

$$\mathrm{Reg}(T) = \sum_{t=1}^{t} \sup_{x \in \mathcal{X}_t} \left( \langle x, \theta^\star \rangle - \langle X_t, \theta^\star \rangle \right).$$

A common approach to tackle the linear bandit problem is to follow an *upper confidence bound* (UCB) strategy. This involves the following protocol. At each round $t$, we first derive a confidence set $\mathcal{C}_{t-1}$, based on the arm-reward pairs $(X_s, Y_s)_{s \leq t-1}$. This is a random set (as it depends on the past noise realisations), which must be constructed ensuring that $\theta^\star \in \mathcal{C}_{t-1}$ with high probability. More precisely, the regret can be effectively controlled if one can ensure that $\theta^\star$ uniformly belongs to every set $(\mathcal{C}_t)_{t \geq 1}$, with high probability (a property often referred to as *anytime validity*). Then, for every available arm $x$, we let

$$\mathrm{UCB}_{\mathcal{C}_{t-1}}(x) = \max_{\theta \in \mathcal{C}_{t-1}} \langle x, \theta \rangle.$$

By definition, this is a high-probability upper bound on $\langle x, \theta^\star \rangle$, which justifies the name "upper confidence bound". The idea is then to *optimistically* pick as $X_t \in \mathcal{X}_t$ the arm maximising $\mathrm{UCB}_{\mathcal{C}_{t-1}}$.

A key technical challenge in designing a UCB algorithm is to construct the anytime valid confidence sequence $(\mathcal{C}_t)_{t \geq 1}$. Typically, under sub-Gaussian assumptions on the noise, these sets take the form of an ellipsoid, centred on a (regularised) maximum likelihood estimator. Explicitly, we often have

$$\mathcal{C}_t = \left\{ \theta \in \Theta \ : \ \|\theta - \widehat{\theta}_t\|_{V_t}^2 \leq \beta_t^2 \right\},$$

where $\widehat{\theta}_t$ is the least-squares estimator of $\theta^\star$, $V_t$ is the *feature-covariance* matrix and $\beta_t$ is a radius carefully chosen so that the high-probability coverage requirement is satisfied. In this work, to construct the confidence sets we will leverage an *online-to-confidence-set-conversion* approach, a method that reduces the problem of proving statistical concentration bounds to proving existence of well-performing algorithms for an associated game of *sequential probability assignment*. We refer to Section 4 for more details on our technique to construct the confidence sequence.

## 3 Linear bandits with non-i.i.d. observation noise

We study a variant of the standard linear stochastic bandit problem where the observation-noise variables feature dependencies across different rounds. We focus on the case of weakly stationary noise, meaning we assume all the $\varepsilon_t$ to have the same marginal distribution. However, the core assumption we make is what we call *mixing sub-Gaussianity*. This provides a way to control how dependencies decay as the time between two observations increases. It is defined in terms of a sequence of mixing coefficients $\phi_d$, which quantify this decay.

**Assumption 1** (Mixing sub-Gaussianity). *Fix $\sigma > 0$ and let $\phi = (\phi_d)_{d \geq 0}$ be a non-negative and non-increasing sequence. We say that the random sequence $(\epsilon_t)_{t \geq 1}$ is $(\sigma, \phi)$-mixing sub-Gaussian if*

---

[1]We remark that, more generally, one can consider the case where the $X_t$ as well are randomised, namely contain additional randomness that is not included in the noise. To take this into account, one can add this other source or randomness in the filtration. However, since in our case we will only consider a non-randomised bandit algorithm, we omit this to simplify our analysis.

124  $\varepsilon_t$ *is centred and $\sigma$-sub-Gaussian for every $t$, and, for all $d \geq 0$ and all $t > d$, we have*

$$\left| \mathbb{E}\left[\epsilon_t \,|\, \mathcal{F}_{t-d}\right] \right| \leq \phi_d \tag{1}$$

125  *and*

$$\mathbb{E}\left[\exp \lambda(\epsilon_t - \mathbb{E}\left[\epsilon_t \,|\, \mathcal{F}_{t-d}\right]) \,|\, \mathcal{F}_{t-d}\right] \leq e^{\frac{\lambda^2 \sigma^2}{2}}, \qquad \forall \lambda > 0. \tag{2}$$

126  Clearly, the above assumption generalises the standard conditionally sub-Gaussian assumption (that
127  can be recovered by setting $\phi_d = 0$ for all $t$), sometimes considered in the bandit literature. Although
128  this might look like an unusual mixing assumption, it is very natural for our problem at hand, and
129  can be weaker than standard mixing hypotheses. For instance, if the noise sequence is $\varphi$-mixing
130  (see Bradley, 2005) and each $\varepsilon_t$ is centred and bounded in $[-a, b]$, it is straightforward to check that
131  $|\mathbb{E}[\varepsilon_t | \mathcal{F}_{t-d}]| \leq (a + b)\phi_d$, and so Assumption 1 is satisfied since the boundedness automatically
132  implies sub-Gaussianity. In the rest of the paper we assume $\sigma = 1$ for simplicity.

133  Under Assumption 1, we can build the confidence sequence needed for our UCB algorithm. We state
134  this result below, but defer the explicit derivation to Section 4 (see Corollary 1 there).

**Proposition 1.** *For some given $\phi$, let the noise satisfy Assumption 1 with $\sigma = 1$. Fix $\delta \in (0, 1)$, $\lambda > 0$, and $d \geq 1$. For $t \geq 1$ let*

$$\mathcal{C}_t = \left\{ \theta \in \mathcal{B}(B) \,:\, \tfrac{1}{2}\|\theta - \widehat{\theta}_t\|_{V_t}^2 \leq \tfrac{dp}{2} \log \frac{(B+1)^2 e \max(dp, t+d)}{dp} + 2\lambda B^2 + t\phi_d(B + 1) + d \log \tfrac{d}{\delta} \right\},$$

*where $V_t = \sum_{s=1}^{t} X_t X_t^\top + \lambda \mathrm{Id}$, and $\widehat{\theta}_t = \arg\min_{\theta \in \mathcal{B}(B)} \sum_{s=1}^{n}(\langle \theta, X_t \rangle - Y_t)^2$. Then, $(\mathcal{C}_t)_{t \geq 1}$ is an anytime valid confidence sequence, in the sense that*

$$\mathbb{P}\left(\theta^\star \in \mathcal{C}_t, \, \forall t \geq 1\right) \geq 1 - \delta.$$

135  Leveraging the confidence sequence above, we can define a UCB approach for our problem (Algo-
136  rithm 1). At a high level, the algorithm operates by taking the confidence sets defined in Proposition
137  1, and selecting the arm optimistically, as in the standard UCB. A key point is that a delay $d$ is
138  introduced, which at round $t$ restricts the agent to use only the information available from the first
139  $t - d$ rounds. Although the actual technical reason behind this restriction will become fully clear only
140  with the analysis of the coming sections, one can intuitively think of it as a way to prevent overfitting
141  to recent noise, which might be highly correlated. If $d$ is sufficiently large, the noise observed in
142  each round $t$ will be sufficiently decorrelated from the previous observations, which allows accurate
143  estimation and uncertainty quantification of the true parameter $\theta^\star$ and the associated rewards.

---

**Algorithm 1** Mixing-LinUCB

set $d > 0$
**for** $i \in \{1, 2, \ldots d\}$ **do**
    play an arbitrary $X_i$ and observe $Y_i$
**end for**
**for** $t \in \{d + 1, \ldots\}$ **do**
    $X_t = \arg\max_{x \in \mathcal{X}_t} \mathrm{UCB}_{\mathcal{C}_{t-d}}(x)$, where $\mathcal{C}_{t-d}$ is as in Proposition 1
    play $X_t$ and observe reward $Y_t$
**end for**

---

144  In Section 5 we provide a detailed analysis of the regret of the algorithm that we proposed. For
145  instance, assuming that the mixing coefficients decay exponentially as $\phi_d = Ce^{-d/\tau}$ (*geometric*
146  *mixing*), we show that the regret can be upper bounded in high probability as

$$\mathrm{Reg}(T) \leq \mathcal{O}\left(\tau p\sqrt{T}\log(T)^2 + \tau \log T \sqrt{pT \log T}\right).$$

147  We refer to Theorem 2 and Corollary 2 in Section 5 for more details.

## 4  Constructing the confidence sequence

149  In this section we derive a confidence sequence for linear models with non-i.i.d. noise. First, we
150  briefly describe the online-to-confidence-set conversion scheme from Clerico et al. (2025), which
151  serves as our starting point. We then extend this technique to handle mixing noise.

## 4.1 Online-to-confidence set conversion for i.i.d. data

Before proceeding for the analysis of mixing sub-Gaussian noise, which is the focus of this work, we start by describing how to derive a confidence sequence when the noise is independent (or conditionally) centred and sub-Gaussian across different rounds, as in Clerico et al. (2025). The online-to-confidence sets framework that we consider instantiates an abstract game played between an *online learner* and an *environment*. We define the squared loss $\ell_s(\theta) = \frac{1}{2}(\langle \theta, X_s \rangle - Y_s)^2$. For each round $s = 1, \ldots, t$, the following steps are repeated:

1. the environment reveals $X_s$ to the learner;

2. the learner plays a distribution $Q_s \in \Delta_{\mathbb{R}^p}$;

3. the environment reveals $Y_s$ to the learner;

4. the learner suffers the log loss $\mathcal{L}_s(Q_s) = -\log \int_{\mathbb{R}^p} \exp(-\ell_s(\theta)) \mathrm{d}Q_s(\theta)$.

This game is a special case of a well-studied problem called *sequential probability assignment* (Cesa-Bianchi and Lugosi, 2006). The learner can use any strategy to choose $Q_1, \ldots, Q_t$, as long as each $Q_s$ depends only on $X_1, Y_1, \ldots, X_{s-1}, Y_{s-1}, X_s$. We define the *regret* of the learner against a (possibly data-dependent) comparator $\bar{\theta} \in \mathbb{R}^p$ as

$$\mathrm{Regret}_t(\bar{\theta}) = \sum_{s=1}^t \mathcal{L}_s(Q_s) - \sum_{s=1}^t \ell_s(\bar{\theta}) \,.$$

Clerico et al. (2025) provide a regret bound upper bound (Proposition 3.1 there) for when the learner's strategy is from an *exponential weighted average* (EWA) forecaster with a centred Gaussian prior $Q_1$. However, to account for the presence of dependencies in our analysis, we will need the prior's support to be bounded. We hence state here a regret bound (whose proof is deferred to Appendix A.2) for the regret of an EWA forecaster with a uniform prior.

**Proposition 2.** *Fix $B > 0$ and consider the EWA forecaster with as prior the uniform distribution on $\mathcal{B}(B + 1)$. Then, for all $\bar{\theta} \in \mathcal{B}(B)$ and any $t \geq 1$,*

$$\mathrm{Regret}_t(\bar{\theta}) \leq \frac{p}{2} \log \frac{(B + 1)^2 e \max(p, t)}{p} \,.$$

We remark that, by adding and subtracting the total log loss of the learner, the excess loss of $\theta^\star$ (relative to $\bar{\theta}$) can be rewritten as

$$\sum_{s=1}^t \ell_s(\theta^\star) - \sum_{s=1}^t \ell_s(\bar{\theta}) = \mathrm{Regret}_t(\bar{\theta}) + \sum_{s=1}^t \ell_s(\theta^\star) - \sum_{s=1}^t \mathcal{L}_s(Q_s) \,. \tag{3}$$

This simple decomposition is the key idea in the online-to-confidence sets scheme.

Since the noise is conditionally sub-Gaussian and the distributions played by the online learner are predictable ($Q_s$ cannot depend on $Y_s$), $\sum_{s=1}^t \ell_s(\theta^\star) - \sum_{s=1}^t \mathcal{L}_s(Q_s)$ is the logarithm of a non-negative super-martingale (cf. the no-hypercompression inequality in Grünwald, 2007 or Proposition 2.1 in Clerico et al., 2025) with respect to the noise filtration $(\mathcal{F}_t)_{t \geq 1}$.[2] Henceforth, from Ville's inequality (a classical anytime valid Markov-like inequality that holds for non-negative super-martingales) one can easily derive that $\theta^\star \in \mathcal{C}_t$ (uniformly for all $t$) with probability at least $1 - \delta$, where

$$\mathcal{C}_t = \left\{ \theta \in \mathbb{R}^p : \sum_{s=1}^t \ell_s(\theta) - \sum_{s=1}^t \ell_s(\bar{\theta}) \leq \mathrm{Regret}_t(\bar{\theta}) + \log \frac{1}{\delta} \right\} \,.$$

This result can be relaxed by replacing $\mathrm{Regret}_t(\bar{\theta})$ by any known regret upper bound for the online algorithm used in the abstract game (*e.g.*, the bound of Proposition 2 for the EWA forecaster).

---

[2]For simplicity, since this will be the case for our bandit strategy, we assume throughout the paper that $X_t$ is fully determined given the past noise (see footnote 1).

## 4.2 Confidence sequence under mixing sub-Gaussian noise

The standard online-to-confidence sets scheme relies on the fact that $\sum_{s=1}^{t} \ell_s(\theta^\star) - \sum_{s=1}^{t} \mathcal{L}_s(Q_s)$ is the logarithm of a non-negative super-martingale, whose fluctuations can be controlled uniformly in time thanks to Ville's inequality. However, this property hinges on the fact that the noise is assumed to be conditionally centred and sub-Gaussian, which now is not anymore the case. Yet, thanks to our mixing assumption, if we restrict our focus on rounds that are sufficiently far apart, the mutual dependencies get weaker, and the exponential of the sum behaves *almost* like a martingale. This insight suggests to partition the rounds into blocks, whose elements are mutually far apart, then apply concentration results to each block, and finally use a union bound to recover the desired confidence sequence spanning all rounds. We remark that this is a classical approach to derive concentration results for mixing processes, often referred to as the *blocking* technique (Yu, 1994).

In order for the online-to-confidence sets scheme to leverage the blocking strategy outlined above, the abstract online game used for the analysis must be designed in a way that is compatible with the block structure. To address this point, we adopt an approach inspired by Abélès et al. (2025), who introduced delays in the feedbacks received by the online learner in order to address a similar challenge. More precisely, we will now consider the following *delayed-feedback* version of the online game. Fix a delay $d > 0$. For each round $s = 1, \ldots, t$, the following steps are repeated:

1. the environment reveals to the learner $X_s$, which is assumed to be $\mathcal{F}_{s-d}$-measurable;
2. the learner plays a distribution $Q_s \in \Delta_{\mathbb{R}^p}$;
3. if $s > d$, the environment reveals $Y_{s-d+1}$ to the learner;
4. the learner suffers the log loss $\mathcal{L}_s(Q_s) = -\log \int_{\mathbb{R}^p} \exp(-\ell_s(\theta)) \mathrm{d}Q_s(\theta)$.

Note that the delay $d$ only applies for the rewards, while $Q_s$ can still depend on $X_s$. Indeed, the choice of $X_s$ in our mixing UCB algorithm is already "delayed", as it depends on $\mathcal{C}_{t-d}$ (see Algorithm 1).

Of course, in this setting the decomposition of (3) is still valid. We now want to deal with the concentration of $\sum_{s=1}^{t} \ell_s(\theta^\star) - \sum_{s=1}^{t} \mathcal{L}_s(Q_s)$ via the blocking technique. For convenience, let us write $D_t = \ell_t(\theta^\star) - \mathcal{L}_t(Q_t)$. We denote as $S^{(i)} = (S_k^{(i)})_{k \geq 1}$ the subsequence defined as $S_k^{(i)} = \sum_{j=1}^{k} D_{i+(j-1)d}$. The key idea is now that each of these $S^{(i)}$ behaves as the log of a martingale, up to a cumulative remainder that accounts for the conditional mean shift in the mixing sub-Gaussianity assumption. In particular, Ville's inequality and a union bound yield the following.

**Lemma 1.** *Fix a delay $d > 0$ and $\delta \in (0,1)$. We have that*

$$\mathbb{P}\left( \sum_{s=1}^{t} \left( \ell_s(\theta^\star) - \mathcal{L}_s(Q_s) \right) \leq t\phi_d B + d \log \frac{d}{\delta}, \ \forall t \geq 1 \right) \geq 1 - \delta.$$

Now that we have a concentration result to control $S_t$, we only need to be able to upper bound the regret of an algorithm for the "delayed" online game that we are considering. To this purpose, we propose the following approach. We run $d$ independent EWA forecaster (with uniform prior), each one only making prediction and receiving the feedback once every $d$ rounds. More explicitly, the first forecaster acts at rounds $1, d+1, 2d+1...$, the second at round $2, d+2, 2d+2...$, and so on. As a direct consequence of Proposition 2, by summing the individual regret upper bounds we get a regret bound for the joint forecaster, which at each round returns the distribution predicted by the currently active forecaster. This technique of partitioning rounds into blocks for the regret analysis of online learning is common in the literature (*e.g.*, see Weinberger and Ordentlich, 2002).

**Lemma 2.** *Fix $B > 0$, $d > 0$, and consider a strategy with $d$ independent EWA forecasters outlined above, all initialised with the uniform distribution on $\mathcal{B}(B+1)$ as prior. For all $\bar{\theta} \in \mathcal{B}(B)$ and $t \geq 1$,*

$$\mathrm{Regret}_t(\bar{\theta}) \leq \frac{dp}{2} \log \frac{(B+1)^2 e \max(dp, t+d)}{dp}.$$

Putting together what we have, we get a confidence sequence suitable for our mixing UCB algorithm.

**Theorem 1.** *Consider the setting introduced above. Fix $\delta \in (0,1)$ and a delay $d > 0$. Assume as known that $\theta^\star \in \mathcal{B}(B)$. Let $\widehat{\theta}_t = \arg\min_{\theta \in \mathcal{B}(B)} \{\sum_{s=1}^{t} \ell_s(\theta)\}$ and $\Lambda_t = \sum_{s=1}^{t} X_s X_s^\top$. Define*

$$\mathcal{C}_t = \left\{ \theta \in \mathcal{B}(B) : \frac{1}{2} \|\theta - \widehat{\theta}_t\|_{\Lambda_t}^2 \leq \frac{dp}{2} \log \frac{(B+1)^2 e \max(dp, t+d)}{dp} + t\phi_d(B+1) + d \log \frac{d}{\delta} \right\}.$$

*Then, $(\mathcal{C}_t)_{t\geq 1}$ is an anytime valid confidence sequence for $\theta^\star$, namely*

$$\mathbb{P}\big(\theta^\star \in \mathcal{C}_t\,,\ \forall t \geq 1\big) \leq 1 - \delta\,.$$

*Proof.* The optimality of $\widehat{\theta}_t$ implies $\sum_{s=1}^t \langle \theta - \widehat{\theta}_t, \nabla \ell_s(\widehat{\theta}_t)\rangle \geq 0$, for all $\theta \in \mathcal{B}(B)$. As $\sum_{s=1}^t \ell_s$ is quadratic, it equals its second order Taylor expansion around $\widehat{\theta}_t$ and its Hessian is everywhere $\Lambda_t$. So,

$$\frac{1}{2}\|\theta - \widehat{\theta}_t\|_{\Lambda_t}^2 \leq \frac{1}{2}\|\theta - \widehat{\theta}_t\|_{\Lambda_t}^2 + \sum_{s=1}^t \big\langle \theta - \widehat{\theta}_t, \nabla \ell_s(\widehat{\theta}_t)\big\rangle = \sum_{s=1}^t \big(\ell_s(\theta) - \ell_s(\widehat{\theta}_t)\big)\,,$$

for any $\theta \in \mathcal{B}(B)$. This, together with (3), Lemma 1, and Lemma 2, yields the conclusion. $\qquad\square$

We remark that the confidence sets of Theorem 1 take the form of the intersection between the ball $\mathcal{B}(B)$ and the "ellipsoid" $\{\theta : \|\theta - \widehat{\theta}_t\|_{\Lambda_t} \leq \beta_t\}$, for a suitable radius $\beta_t$. In order to implement and analyse the bandit algorithm, it will be more convenient to work with a relaxation of these sets, a pure ellipsoid not intersected with $\mathcal{B}(B)$. We make this explicit in the following corollary.

**Corollary 1.** *Fix $\lambda > 0$, $d > 0$, and $\delta \in (0,1)$. For $t \geq 1$, let $V_t = \Lambda_t + \lambda\mathrm{Id}$. Assuming that $\theta^\star \in \mathcal{B}(B)$, the following compact ellipsoids define an anytime valid confidence sequence for $\theta^\star$:*

$$\mathcal{C}_t = \left\{\theta \in \mathcal{B}(B)\ :\ \tfrac{1}{2}\|\theta - \widehat{\theta}_t\|_{V_t}^2 \leq \tfrac{dp}{2}\log\frac{(B+1)^2 e \max(dp, t+d)}{dp} + 2\lambda B^2 + t\phi_d(B+1) + d\log\tfrac{d}{\delta}\right\}.$$

*Proof.* Let $\beta_t^2 = dp\log\frac{(B+1)^2 e \max(dp, t+d)}{dp} + 2t\phi_d(B+1) + 2d\log\frac{d}{\delta}$. From Theorem 1, with probability at least $1 - \delta$, uniformly for every $t$, $\|\theta^\star - \widehat{\theta}_t\|_{\Lambda_t}^2 \leq \beta_t^2$. Adding to both sides of this inequality $\frac{\lambda}{2}\|\theta^\star - \widehat{\theta}_t\|_2^2$, and relaxing the RHS using that $\|\theta^\star - \widehat{\theta}_t\|_2^2 \leq 4B^2$, we conclude. $\qquad\square$

# 5 Regret bounds for Mixing-LinUCB

In this section, we establish worst-case and gap-dependent cumulative regret bounds for mixing UCB algorithm (Mixing Lin-UCB). However, to account for the fact that Mixing-LinUCB selects actions with delays, the standard elliptical potential arguments must be modified. Throughout this section, we let $R_t = \langle \theta^\star, X_t^\star - X_t\rangle$ (where $X_t^\star = \arg\max_{x\in\mathcal{X}_t}\langle\theta^\star, x\rangle$) denote the regret in round $t$, and $\beta_t^2 = dp\log\frac{(B+1)^2 e \max(dp, t+d)}{dp} + 4\lambda B^2 + 2t\phi_d(B+1) + 2d\log\frac{d}{\delta}$ denote the squared radius of the ellipsoid $\mathcal{C}_t$ in Corollary 1.

## 5.1 Worst-case regret bounds

First, following the regret analysis in Abbasi-Yadkori et al. (2011) (see also Section 19.3 in Lattimore and Szepesvári, 2020), we upper bound the instantaneous regret. From our boundedness assumptions ($\theta^\star \in \mathcal{B}(B)$ and $\mathcal{X}_t \subseteq \mathcal{B}(1)$), we easily deduce that $R_t \leq 2B$. Under the event that our confidence sequence contains $\theta^\star$ at every step $t$, we have another bound on $R_t$. If we define $\widetilde{\theta}_{t-d} \in \mathcal{C}_{t-d}$ to be the point at which $\langle\widetilde{\theta}_{t-d}, X_t\rangle = \mathrm{UCB}_{\mathcal{C}_{t-d}}(X_t)$, then from the definition of $X_t$ we have

$$\langle\theta^\star, X_t^\star\rangle \leq \max_{x\in\mathcal{X}_t}\max_{\theta\in\mathcal{C}_{t-d}}\langle\theta, x\rangle = \max_{x\in\mathcal{X}_t}\mathrm{UCB}_{\mathcal{C}_{t-d}}(x) = \mathrm{UCB}_{\mathcal{C}_{t-d}}(X_t) = \langle\widetilde{\theta}_{t-d}, X_t\rangle\,.$$

Recall that, for all $s$, $V_s = \Lambda_s + \lambda\mathrm{Id}$, which is invertible as $\lambda > 0$. Thus, by Cauchy-Schwarz,

$$R_t \leq \langle\widetilde{\theta}_{t-d} - \theta^\star, X_t\rangle \leq \|\widetilde{\theta}_{t-d} - \theta^\star\|_{V_{t-d}}\|X_t\|_{V_{t-d}^{-1}} \leq 2\beta_{t-d}\|X_t\|_{V_{t-d}^{-1}}\,.$$

This means that the instantaneous regret satisfies the bound

$$R_t \leq 2\max(B, \beta_{t-d})\min(1, \|X_t\|_{V_{t-d}^{-1}})\,. \tag{4}$$

Next, we separate the regret suffered in the first $d$ rounds and the remaining $T - d$ rounds. We then use Cauchy-Schwarz once more, and the fact that $\beta_t$ is increasing in $t$, to obtain

$$\text{Reg}(T) \leq 2dB + \sqrt{(T-d)\sum_{t=d+1}^T R_t^2}$$
$$\leq 2dB + \sqrt{4(T-d)\max(B^2, \beta_{T-d}^2)\sum_{t=d+1}^T \min(1, \|X_t\|_{V_{t-d}^{-1}}^2)}\,.$$

At this point, we must depart from the standard linear UCB analysis (Abbasi-Yadkori et al., 2011; Lattimore and Szepesvári, 2020). We bound the sum of the *elliptical potentials* $\sum_{t=d+1}^T \min(1, \|X_t\|_{V_{t-d}^{-1}}^2)$ using the following variant of the well-known "elliptical potential lemma" (see Appendix), which accounts for the fact that the feature covariance matrix $V_{t-d}$ is updated with a delay of $d$ steps.

**Lemma 3.** *For all $T \geq 1$,*

$$\sum_{t=d+1}^T \min(1, \|X_t\|_{V_{t-d}^{-1}}^2) \leq 2dp \log(1 + \tfrac{T}{\lambda dp})\,.$$

We can now state a worst-case regret upper bound for Mixing-LinUCB.

**Theorem 2.** *Fix $\lambda = 1/B^2$, $d > 0$ and $\delta \in (0,1)$. With probability at least $1 - \delta$, for all $T > d$, the regret of Mixing-LinUCB satisfies*

$$\text{Reg}(T) \leq 2dB + \sqrt{8dpT\max(B^2, \beta_T^2)\log(1 + \tfrac{B^2 T}{dp})}\,.$$

From the definition of $\beta_T$, we see that this regret bound is of the order

$$\text{Reg}(T) = \mathcal{O}\left(dB + dp\sqrt{T}\log\tfrac{TB}{dp} + T\sqrt{Bdp\phi_d \log\tfrac{TB}{dp}} + d\sqrt{pT\log\tfrac{TB}{p\delta}}\right)\,.$$

For any fixed (*i.e.*, not depending on $T$) delay $d$, this regret bound is linear in $T$. To obtain meaningful regret bounds, it is therefore crucial to set $d$ as a function of $T$ and the rate at which the mixing coefficients decay to zero[3]. Under the assumption that the noise variables are either geometrically or algebraically mixing, we obtain the following worst-case regret bounds.

**Corollary 2.** *Suppose that the noise satisfies Assumption 1 with $\phi_d = Ce^{-\frac{d}{\tau}}$ for some $C, \tau > 0$ (geometric mixing), and set $d = \lceil \tau \log \tfrac{BCT}{p} \rceil$. Then, the regret of Mixing-LinUCB satisfies*

$$\text{Reg}(T) = \mathcal{O}\left(\tau p\sqrt{T}\left(\log\tfrac{TB\max(1,C)}{p}\right)^2 + p\sqrt{T}\tau\log\tfrac{TB\max(1,C)}{p} + \tau\log\tfrac{BCT}{p}\sqrt{pT\log\tfrac{TB}{p\delta}}\right)\,.$$

**Corollary 3.** *Suppose that the noise satisfies Assumption 1 with $\phi_d = Cd^{-r}$ for some $C > 0$ and $r > 0$ (algebraic mixing), and set $d = \lceil CT^{1/(1+r)} \rceil$. Then, the regret of Mixing-LinUCB satisfies*

$$\text{Reg}(T) \leq \mathcal{O}\left(CBT^{1/(1+r)} + T^{\frac{3+r}{2(1+r)}}\left(Cp\log\tfrac{TB}{dp} + C\sqrt{Bp\log\tfrac{T^{r/(1+r)}B}{Cp}} + \sqrt{p\log\tfrac{TB}{p\delta}}\right)\right)\,.$$

Up to a factor of $\tau \log T$, the bound for geometrically mixing noise matches the regret bound for linear UCB with i.i.d. noise. This bound is trivial for $r \leq 1$, however for $r > 1$ we get sublinear regret, and in particular we recover standard rates up to logarithmic factors in the limit where $r \to \infty$.

## 5.2 Gap-dependent regret bounds

Under the assumption that, each round, the gap between the expected reward of the optimal arm and the expected reward of any other arm is at least $\Delta > 0$, we get regret bounds with better dependence

---

[3]If $T$ is unknown, one could probably use doubling tricks to set the value of $d$, but we do not pursue this here.

on $T$. More precisely, define the *minimum gap* $\Delta = \min_{t \in [T]} \min_{x \in \mathcal{X}_t : x \neq X_t^\star} \langle X_t^\star - x, \theta^\star \rangle$, and assume that $\Delta > 0$. Since we either have $R_t = 0$ or $R_t \geq \Delta > 0$, it follows that

$$R_t \leq R_t^2 / \Delta \,.$$

In our worst-case analysis, we showed that

$$\sum_{t=d+1}^{T} R_t^2 \leq 8dp \max(B^2, \beta_T^2) \log(1 + \tfrac{T}{\lambda dp}) \,.$$

Combined with the previous inequality, we obtain the following gap-dependent regret bound.

**Theorem 3.** *Fix $\lambda = 1/B^2$, $d > 0$, and $\delta \in (0, 1)$. With probability at least $1 - \delta$, for all $T > d$, the regret of Mixing-LinUCB satisfies*

$$\operatorname{Reg}(T) \leq 2dB + \frac{8dp}{\Delta} \max(B^2, \beta_T^2) \log \left( 1 + \frac{B^2 T}{dp} \right) \,.$$

Similarly to the worst-case bound in Theorem 2, for any fixed $d > 0$, this regret bound is linear in $T$. By setting $d$ as a suitable function of $T$, we obtain the following gap-dependent regret bounds under geometrically or algebraically mixing noise.

**Corollary 4.** *Suppose that the noise variables are geometrically mixing and set $d = \lceil \tau \log \frac{BCT}{p} \rceil$. Then the regret of Mixing-LinUCB satisfies*

$$\operatorname{Reg}(T) = \mathcal{O} \left( \frac{8\tau p}{\Delta} \left( log \frac{BCT}{p} \right)^2 \log \left( 1 + \frac{B^2 T}{p\tau \log \frac{BCT}{p}} \right) \left( \frac{p}{2} \log \frac{T}{p\tau} + \log \frac{\tau \log \frac{BCT}{p}}{\delta} \right) \right) \,.$$

**Corollary 5.** *Suppose that the noise variables are algebraically mixing and set $d = \lceil CT^{1/(1+r)} \rceil$. Then the regret of Mixing-LinUCB satisfies*

$$\operatorname{Reg}(T) = \mathcal{O} \left( \frac{8Cp}{\Delta} T^{\frac{2}{1+r}} \log \left( 1 + \frac{B^2 T}{pCT^{1/(1+r)}} \right) \left( \frac{p}{2} \log \frac{T}{p\tau} + \log \frac{CT^{1/(1+r)}}{\delta} \right) \right) \,.$$

## 6  Conclusion

We leave several interesting questions open for future research. Some of these are listed below.

An important limitation of our algorithm is that it requires the knowledge of the mixing coefficients (or at least an upper-bound on them). It would be interesting to explore the possibility of relaxing this assumption and to design an algorithm which infers the mixing coefficients while minimizing the regret. We note that the problem of estimating mixing coefficients is already a hard problem on its own right, with tight sample-complexity results only available in special cases such as Markov chains (Hsu et al., 2019; Wolfer, 2020). We also note that in order to recover the standard rate for the regret bound, the delay $d$ introduced in our algorithm need to be chosen as a function of the horizon $T$. We believe that this could be fixed at little conceptual expense by using time-varying delay in the analysis, but we did not attempt to work out the (potentially non-trivial) details here.

Another limitation is that our analysis assumed throughout that the adversary picking the decision sets $\mathcal{X}_t$ is oblivious, which is typically not required in linear bandit problems. For us, this was necessary to avoid potential statistical dependence between decision sets and the nonstationary observations. We believe that this issue can be handled at least for some classes of adversaries. For instance, it is easy to see that our analysis would carry through under the assumption that the decision sets be selected based on delayed information only. We leave the investigation of this question under more realistic assumptions open for future work.

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
