# OpenReview forum: "Linear Bandits with Non-i.i.d. Noise"
_NeurIPS.cc/2025/Conference — Submitted to NeurIPS 2025_

### Official Review · Reviewer_svgi · 2025-06-27

**Clarity:** 3
**Significance:** 2
**Originality:** 2
**Rating:** 4
**Confidence:** 4

**Summary:**

This paper addresses the linear stochastic bandit problem by relaxing the standard i.i.d. assumption for observation noise, instead allowing for sub-Gaussian noise with dependencies that decay over time. The authors introduce a concept called "mixing sub-Gaussianity" to formalize this setting and develop a new UCB-style algorithm, Mixing-LinUCB, which introduces a delay to manage these dependencies. By adapting the online-to-confidence-set conversion scheme with a blocking technique, the paper derives regret bounds for the algorithm that are expressed in terms of the dependency decay rate. A key result is that for geometrically mixing noise, these bounds can recover standard regret rates, with an additional factor related to the mixing time.

**Questions:**

1. Novelty: While the paper addresses the important problem of linear bandits with non-i.i.d. noise, the core framework appears to be a novel and effective synthesis of several existing techniques. Specifically, it combines the online-to-confidence-set conversion scheme , the classic "blocking technique" for mixing processes , and the use of "delayed feedback" in an abstract online game to accommodate the block structure, which has been explored in other contexts. Could the authors elaborate on the specific technical novelty required to make these components work together for the linear bandit problem?

2. Sensitivity to $d$: The derived regret bounds depend critically on the choice of the delay parameter $d$. Could the authors provide more insight into the algorithm's sensitivity to the choice of d? A simple numerical experiment, even on synthetic data, would be highly valuable to address this.

**Ethical Concerns:**

["NO or VERY MINOR ethics concerns only"]

**Final Justification:**

The authors have addressed my concerns, and I would like to keep my original score.

**Limitations:**

yes

**Quality:**

3

**Strengths And Weaknesses:**

- Strengths
	- Significant and Well-Motivated Problem: The paper tackles a crucial limitation of most bandit literature. The standard i.i.d. noise assumption is often unrealistic in practical applications such as online advertising or recommendation systems, where external factors can introduce temporal correlations in user feedback. This work provides a principled approach to model and handle such dependencies, significantly increasing the practical relevance of the linear bandit framework.
	- Novel Technical Methodology with Strong Theoretical Guarantees: The core strength of this paper lies in its innovative and sound theoretical analysis. The authors successfully adapt the modern online-to-confidence-set conversion framework to handle dependent data by introducing a novel blocking technique within a delayed abstract game. This powerful machinery leads to compelling and rigorous regret bounds that explicitly depend on the noise's mixing rate. Crucially, the analysis demonstrates that for geometrically mixing noise, the algorithm achieves nearly optimal regret, recovering the standard rates up to logarithmic factors and the mixing time.

- Weaknesses
	- I find Assumption 1 somewhat unintuitive. It seems more natural to assume that each $\epsilon_t$ satisfies a delay-dependent variance bound $\sigma_d$, rather than a magnitude bound $\phi_d$. Could you elaborate on the motivation behind the current assumption and clarify why it is preferred in your analysis?
	- Required Knowledge of Noise Structure: A related and arguably stronger assumption is that selecting the optimal delay $d$ requires a priori knowledge of the mixing coefficients that characterize the noise process (e.g., the decay rate $\tau$ and constant $C$ for geometric mixing, or $r$ for algebraic mixing). The authors themselves acknowledge this limitation, noting that estimating these coefficients is a hard problem in its own right. This creates a methodological tension: the paper solves the problem of unknown noise correlation by assuming the structure of that correlation is known, which can be an equally restrictive assumption in practice.

---

> ### Author Rebuttal · Authors · 2025-07-30
>
> Thank you for your careful reading of our paper and your thoughtful comments! We are particularly glad that you have found our problem setting interesting and have appreciated the rigor of our theoretical results. We respond to your questions below.
>
> Intuition about the assumption: Note that our assumption corresponds to bounding the *bias* rather than the variance of the noise, stating that the conditional bias of the observation in time $t$ approaches zero when conditioning on the history up to round $t-d$ for large values of $d$. Intuitively, this amounts to requiring that the observation noise becomes uncorrelated when considering time steps that are “far enough” apart. The constant $\sigma$ roughly corresponds to the conditional variance of the observation noise, which is assumed to remain fixed. In our view, enforcing a decaying time-dependent bound on the variance would be somewhat less interesting, as that would allow all observations to be unbiased and would imply that all observations become nearly noiseless after a sufficiently large delay. Your suggestion for such an alternative assumption is interesting nevertheless, and we will be happy to include a more detailed discussion about this in the final version.
>
> While we definitely agree that requiring the mixing properties of the noise process to be known is a non-trivial assumption, we respectfully disagree that it would be “equally restrictive” as knowing the exact correlations between the noise. Knowing the correlations would enable us to design *unbiased* estimates of the rewards, which would significantly facilitate both the design and the analysis of algorithms. We additionally believe that knowledge of the exact correlation structure of the noise realizations would almost never be available in applications, whereas it is often possible to verify upper bounds on the mixing rates based on prior knowledge of the problem to be solved (e.g., by verifying that the noise behaves roughly like an autoregressive process with bounded parameters). We will provide a more detailed discussion of this in the final version.
>
> Regarding your further questions:
>
> **Novelty:** Regarding the technical content of the paper, our main contributions are: 1) a new and natural modeling assumption that extends the classical i.i.d. assumption in bandit theory and captures a new class of dependence structure between data points and 2) an analytic framework inspired by the online-to-confidence-set-conversion scheme that works seamlessly with the newly proposed dependence model. We believe that the resulting concentration inequality and its proof are valuable contributions in their own right, and that the simplicity of the techniques is more of a merit of our work than a drawback. We are not aware of any other comparable concentration result that would have been suitable for addressing the bandit problem we have studied here, thus we feel that there can be no doubts about the novelty of our contribution.
>
> **Sensitivity to $d$:** Our confidence sets and regret bounds are indeed affected by the choice of $d$. This is due to a trade-off mechanism. On one hand, one wants to pick a small $d$ to get an effective dataset size of $t/d$ for the concentration argument. On the other hand, picking a larger $d$ typically ensures that the overall bias is smaller, as $\phi_d$ decays with $d$. In practice, as discussed in Section 5, we believe that the safest way to pick $d$ is to consider the order magnitude of the regret upper-bound. This can be done for the algebraic and geometric mixing (see Corollaries 2 and 3) and the discussion above.

---

> > ### Comment · Reviewer_svgi · 2025-08-09
> >
> > Thank you for your response. I am fine with it, and I would like to discuss it with the other reviewers before making my final recommendation.

---

### Official Review · Reviewer_Uro6 · 2025-06-30

**Clarity:** 4
**Significance:** 2
**Originality:** 3
**Rating:** 2
**Confidence:** 3

**Summary:**

This paper studies the classical linear contextual bandits problem, but relaxes the usual assumption that the noise​ is conditionally i.i.d. sub-Gaussian. Instead, the authors proposes a new assumption that the noise sequence to be mixing sub-Gaussian, and shows that this assumption is not stronger than standard mixing assumptions. Under this assumption, the author proposes mixing-LinUCB algorithm and shows regret bounds for it. The author shows that mixing-LinUCB matches the regret bound up to logarithmic factors in the standard setting.

**Questions:**

- As I mentioned in the above section, I feel like Proposition 1 is almost the same as Theorem 1, so it might be good to rename them the same instead of duplicating.
- Also as I mentioned, I think it would be good to see at least a simulation study to show that your algorithm empirically is comparable to existing approaches, e.g. LinTS, under i.i.d. noise settings.
- Related to the above point, I'd love to see if there are any benchmarks for linear bandits under non-i.i.d. noise and how your algorithm compares to them.

**Ethical Concerns:**

["NO or VERY MINOR ethics concerns only"]

**Final Justification:**

The rebuttal from the authors is generally clear and has solved some of my questions. However, I will keep my score given that my follow-up on either a lower bound demonstrating the optimality of this algorithm or some simple experiments to demonstrate the performance of their algorithm compared to others (or at least get an intuition that it can be much better than others) still remains unsolved.

**Limitations:**

Yes.

**Paper Formatting Concerns:**

No major formatting issues.

**Quality:**

2

**Strengths And Weaknesses:**

Strengths:
- The paper is theoretically solid. Most lemmas and theorems have proofs and the author introduces and explains each result in detail.
- The paper is generally clear. It explains all the concepts well, starting from the definition of linear bandits to the idea of optimism under uncertainty. I think it is decently accessible to general readers who are not expert in bandits.

Weaknesses:
- There is no experiment. Given the description of the algorithm it does not seem too hard to construct at least a simulation (I could be wrong). Also, it would be good to demonstrate that the algorithm works empirically to further show the validity of it. For example, one could simulate a usual linear bandit instance (e.g., in "On Elimination Strategies for Bandit Fixed-Confidence Identification" by Tirinzoni and Degenne) where the noise follows an AR(1) process or some other dependency structures that satisfy the mixing sub-Gaussianity assumption in this paper. One can adjust the dependency level and see how much Mixing-LinUCB outperforms other benchmarks when the noise is very dependent or almost independent. Although I'm not sure what the result would be, seeing big improvements with dependent noise demonstrates that we need new algorithms in this setting instead of just using current ones like LinUCB and LinTS.
- The paper seems hastily written for some parts. For example, Proposition 1 is almost the same as Theorem 1, so I’m not sure why the author named this a separate proposition. Also, the proof of Proposition 1 is missing, and this proposition actually seems quite critical since the algorithm builds on this.
- I’m a bit struggled to find the technical novelty in the proof. In particular, I feel like a lot of it is a reproof of LinUCB and EWA forecaster, which both seem present in existing works. The main novelty I found might be the new notion of mixing sub-Gaussianity. I think the authors could make it more clear about their technical contributions especially given that this is a pure theoretical paper.

---

> ### Author Rebuttal · Authors · 2025-07-30
>
> Thank you for your careful reading of the paper and your constructive remarks!
>
> We would like to highlight that the main contribution of this work is theoretical, and we believe that our results match the standards of the existing literature on bandit problems. In particular, it is common in this literature to focus entirely on theory. As such, we feel that our work is strong enough for publication at NeurIPS even in the absence of experiments.
>
> Thank you for pointing our attention to the small inconsistency about the naming of Proposition 1! We opted to repeat the statement at the end of Section 4.2 for the convenience of the reader, but we agree that the end result might have ended up looking more confusing than necessary. In the final version, we will remove Corollary 1 and simply refer back to Proposition 1.
>
> Regarding the technical content of the paper, our main contributions are: 1) a new and natural modeling assumption that extends the classical i.i.d. assumption in bandit theory and captures a new class of dependence structure between data points and 2) an analytic framework inspired by the online-to-confidence-set-conversion scheme that works seamlessly with the newly proposed dependence model. We believe that the resulting concentration inequality and its proof are valuable contributions in their own right, and that the simplicity of the techniques is more of a merit of our work than a drawback. We are not aware of any other comparable concentration result that would have been suitable for addressing the bandit problem we have studied here, thus we feel that there can be no doubts about the novelty of our contribution.

---

> > ### Comment · Reviewer_Uro6 · 2025-08-04
> >
> > I thank the authors for their responses that addressed my questions. It still would be good to see either a lower bound demonstrating the optimality of this algorithm or some simple experiments to demonstrate the performance of their algorithm compared to others (or at least get an intuition that it can be much better than others).

---

> > > ### Author Response · Authors · 2025-08-08
> > >
> > > We’re pleased to hear that we have addressed your questions. As we have mentioned in some of our other responses, at the current stage we don't have a lower bound, but we nevertheless conjecture that our bounds are tight. This is supported by the lower bounds for statistical estimation pointed out by reviewer H6s2. We will consider working out the details for the final version of the paper, but we nevertheless believe that our results are already interesting enough in their own right to warrant publication.

---

### Official Review · Reviewer_vXHZ · 2025-07-02

**Clarity:** 4
**Significance:** 3
**Originality:** 3
**Rating:** 5
**Confidence:** 2

**Summary:**

This paper introduces the linear stochastic bandits under non-i.i.d. noise modeled by a mixing process.
It loosens standard assumptions used in previous bandit studies, such as conditionally sub-Gaussian noise.
To handle this setting, the authors develop new confidence sequences that remain valid over time despite the dependence in the noise. They propose a delay-based decision-making strategy that avoids using the most recent observations. The resulting algorithm achieves regret upper bounds that adapt to the degree of noise dependence.

**Questions:**

Q1. Out of academic curiosity, I was wondering whether there are alternative ways to model noise dependence beyond mixing processes.

Q2. To handle the dependency of the noise, the proposed method deliberately discards recent observations. Then, could applying this method to real-world scenarios lead to any unexpected difficulties?

Q3. Is it possible to establish a lower bound in this setting, or at least gain some intuition about how it might behave?

**Ethical Concerns:**

["NO or VERY MINOR ethics concerns only"]

**Final Justification:**

I believe the paper still addresses a meaningful problem, so I will maintain my current score. That said, I am not an expert in this specific area, and my review is based on an educated guess. I hope this context is taken into account.

**Limitations:**

Yes.

**Paper Formatting Concerns:**

No formatting issues in this paper.

**Quality:**

3

**Strengths And Weaknesses:**

**Strengths**

Promising paper! Though I am not deeply familiar with the literature on non-i.i.d. noise in bandits, but from my perspective, this work is well-structured.

S1. The paper tackles the linear bandit problem under dependent (non-i.i.d.) noise. This noble setting more closely reflects real-world scenarios.

S2. They introduce a new confidence set construction tailored for dependent noise, leveraging recent advances in sequential probability assignment.

S3. They provide regret upper bound of their algorithm in terms of the decay rate of mixing coefficient.

S4. Clear writing and intuitive description of their claims.

**Weaknesses**
Overall, I do not have major concerns. Potential concerns, such as the reliance on prior knowledge of mixing coefficient and the assumption of an oblivious environment.—are already acknowledged by the authors as limitations of the work.

W1. While the theoretical contributions are clear, the lack of empirical validation raises concerns about the practical applicability. Why you exempt the numerical experiments?

Typo
1. In line 54-55, Abbasi-Yadkori et al. (2011) -> Abbasi-Yadkori et al. (2012)
2. In line 106, Reg(T) = \sum_{t=1}^{T} \sup ...
3. In Proposition 1. (line 134), $ V_t = \sum_{s=1}^{t} X_s X_s^\top + \lambda I_d$
4. In Proposition 1. (line 134), $\sum_{s=1}^{t} ( < \theta, X_s> - Y_s)^2

Abbasi-Yadkori, Yasin, David Pal, and Csaba Szepesvari. "Online-to-confidence-set conversions and application to sparse stochastic bandits." Artificial Intelligence and Statistics. PMLR, 2012.

---

> ### Author Rebuttal · Authors · 2025-07-30
>
> Thank you for the positive feedback and for considering our work as interesting and clearly presented!
>
> We would like to highlight that the main contribution of this work is theoretical, and we believe that our results match the standards of the existing literature on bandit problems. In particular, it is common in this literature to focus entirely on theory. As such, we feel that our work is strong enough for publication at NeurIPS even in the absence of experiments.
>
> Thank you for pointing out the typos! We will make sure to fix all of them in the revised version of our work.
>
> Regarding your detailed questions:
>
> **Mixing assumption:** Mixing processes are considered to be a very natural way to model noise dependence. We refer to “Basic properties of strong mixing conditions: A survey and some open questions” by Bradley (2005) for detailed survey of mixing assumptions and possible alternatives.
>
> **Discarding recent observations:** Note that recent observations are “discarded” only temporarily, and are eventually incorporated into the parameter estimates and thus all later decisions. This makes our method more data-efficient than the simple alternative of subsampling the decision-making rounds at a fixed rate $d$. We can, however, think of scenarios in more challenging settings where discarding recent observations could be harmful — an obvious example would be that of reinforcement learning in Markov decision processes, where not taking the latest state into account could be very harmful. Luckily this is not an issue in our problem setting.
>
> **Lower bounds:** We agree that proving lower bounds would be an interesting direction of future work, with several non-trivial challenges. Our intuition is that the lower bound should match our upper bound for geometric mixing up to logarithmic factors since the rate for the estimation error of the parameters is already nearly minimax optimal (and the problem of decision making is harder than the estimation problem).

---

> > ### Comment · Reviewer_vXHZ · 2025-08-08
> >
> > Thank you for your thoughtful responses to my questions. I will finalize my decision soon.

---

### Official Review · Reviewer_H6s2 · 2025-07-02

**Clarity:** 3
**Significance:** 2
**Originality:** 2
**Rating:** 3
**Confidence:** 3

**Summary:**

This paper revisits the classical stochastic linear bandit problem under weaker assumptions on the exogenous noise process than studied in prior work. In particular, prior work typically assumes the additive noise process in the observation model to be i.i.d. sub-Gaussian. In contrast, this work allows the noise process to be temporally correlated. To model correlations, the authors introduce the notion of a “mixing sub-Guassian” condition. Under this condition, confidence sets are derived for the unknown parameter $\theta$. Relative to the standard confidence set width, the confidence sets under correlated noise are inflated by a delay term that captures the range of dependencies. Once the confidence sets are derived, the rest of the algorithm follows the standard principle of optimism in the face of uncertainty.
The authors derive regret bounds under different types of assumptions on mixing (geometric mixing and polynomial mixing).

**Questions:**

My main question/comment, as discussed above, is that the result in this paper, while novel, seems to be a rather straight	forward application of the blocking technique that has appeared before in stochastic approximation algorithms for RL.

**Ethical Concerns:**

["NO or VERY MINOR ethics concerns only"]

**Limitations:**

YES

**Quality:**

2

**Strengths And Weaknesses:**

Strengths:
-	This appears to be the first work to provide an analysis of linear bandit algorithms under correlated noise.
-	The strategy of using the “online-to-confidence” set conversion to construct the confidence sets is interesting.
-	The derived regret bounds are consistent with what one would intuitively expect. For instance, under geometric mixing (i.e., when the temporal dependencies are weak), the regret bounds are only a factor of the mixing time away from the corresponding bounds under i.i.d. noise.
Weaknesses and Comments:
-	My main comment and concern is that while the aspect of considering correlated noise might be relatively new for the bandit problem, this issue has been extensively studied in the context of stochastic approximation algorithms for RL. See, for instance, references [R1] and [R2] below that employ the so-called blocking technique (similar to this paper) to deal with correlations.

At a high level, samples that are sufficiently spaced out over time are weakly correlated. As such, one can simply sub-sample the data-stream to first generate a sequence of nearly independent observations and then play the standard algorithm on this sub-sampled sequence. Doing so would recover the same guarantees as in the i.i.d. setting, up to a factor of ‘d’, where ‘d’ is the block length/delay/gap between samples that scales with the mixing time (up to log factors). This basic idea is formalized via a coupling argument in references [R1] and [R2] below. Given that this idea is fairly well-known (both for stochastic approximation and for supervised learning), I am not entirely convinced by the significance of the contribution (despite the differences in the elliptical potential argument to account for delays).

[R1] Adapting to mixing time in stochastic optimization with Markovian data, Dorfman and Levy, ICML 2022.

[R2] Finite time analysis of temporal difference learning with linear function approximation: Tail averaging and regularization, Patil et al., AISTATS 2023.

-	The paper only provides upper bounds, and as such, the tightness of the derived bounds as a function of the rate of mixing remains unclear. In this context, I would encourage the authors to take a look at [R3] below to see if similar ideas can be employed to establish lower bounds for the bandit setting under correlated noise.

[R3] Least squares regression with markovian data: Fundamental limits and algorithms, Nagaraj et al., NeuRIPS 2020.

-	It would be good if the authors can provide a more detailed discussion of how the radius of the confidence set under correlated noise compares with that under i.i.d. noise?

---

> ### Author Rebuttal · Authors · 2025-07-30
>
> Thank you for your detailed reading of our paper and your helpful comments! We respond to your questions below.
>
> While we agree that our algorithm design and analysis makes use of some well-established ideas (most notably the “blocking trick” that you highlighted), we would like to argue that the concrete results we prove in this paper are all non-trivial and novel. The main contribution is addressing the issue of non-stationary noise in an emblematic sequential **decision-making** problem: linear contextual bandits. This decision-making problem is distinct from the works you have highlighted, which focus on sequential **prediction** or **estimation**. The bandit problem we consider here is arguably more challenging due to the partial feedback obtained by the learning agent, which necessitates a more careful analysis. Notably, in the references you provided, the learner does not have to deal with partial feedback, which results in a simpler (although obviously still non-trivial) problem. We will highlight these differences in the final version, in order to contextualize our work better. (In hindsight, we feel that we should have mentioned these works in the first version already — thank you for pointing out this omission!)
>
> We also agree that it might be possible to design a more straightforward algorithm by simply skipping $d$ steps ahead in every decision-making round, but we believe that this approach would be less data-efficient in that it would unnecessarily throw away observations that could otherwise be useful for refining the estimates. In contrast, our approach makes use of all data points. Additionally, we would like to highlight that one of our main contributions is designing a new confidence set construction for linear least-squares estimators under dependent noise, and this contribution might be of independent interest.
>
> Even more concretely, our main contributions are 1) a new and natural modeling assumption that extends the classical i.i.d. assumption in bandit theory and captures a new class of dependence structure between data points and 2) an analytic framework inspired by the online-to-confidence-set-conversion scheme that works seamlessly with the newly proposed dependence model. The fact that this allows us to obtain tight and easy-to-interpret regret bounds with a simple proof technique is, at least in our opinion, a *merit* of our newly proposed model and our analytic framework.
>
> As for your more concrete questions:
>
> **Lower bounds:** We agree that proving lower bounds would be an interesting direction of future work. We have found the reference you provided to be interesting and potentially useful, but it is unclear if these techniques could be adapted to our setting. Our intuition is that the lower bound should match our upper bound for geometric mixing up to logarithmic factors since the rate for the estimation error of the parameters is already nearly minimax optimal (and the problem of decision making is harder than the estimation problem).
>
> **Radius of confidence set:** In the i.i.d. case, we can set $d=1$ and $\phi_1=0$ in the confidence set of Theorem 1. This gives an expression that can be compared with the radius provided in Theorem 4.1 of Clerico, Flynn, Kotłowski and Neu (2025). To our knowledge, this is the tightest confidence set in this form available for sub-Gaussian i.i.d. data. The main difference between our result and theirs is the regret bound for the online algorithm involved in the online-to-confidence conversion. They use an EWA forecaster with a Gaussian prior. In order for the proof of our Lemma 1 to go through, we need to use a prior with bounded support. Despite this additional restriction, for the i.i.d. case we still obtain confidence sets with radii that match the radii of the sets in Clerico, Flynn, Kotłowski and Neu (2025) (i.e., O(p/2log(t/p))). We will add some more comments on this in the final version of the paper.

---

### Decision · Program_Chairs · 2025-09-17

**Decision:**

Reject

**Comment:**

The paper studies linear contextual bandits when reward noise is not i.i.d. over time and proposes a theoretically-driven algorithm that constructs confidence sets using the modern idea of online-to-confidence-set conversion -- reducing concentration for correlated stochastic processes to the existence of good sequential probability assignment strategies. This direction -- moving beyond the standard sub-Gaussian, conditionally independent noise to temporally dependent processes -- is nontrivial, with several reviewers appreciating the novelty of importing sequential prediction tools into confidence sets for sequential observations.

During the review and discussion process, the reviewers engaged actively with the authors to respond to important clarifications requested. One reviewer questioned the novelty of the sequential confidence set route and suggested that a spaced subsampling scheme might yield approximately independent samples, after which classical i.i.d.-based analyses could directly apply; the authors maintained that their approach is novel and advantageous in terms of information use but did not provide a quantitative comparison (rates, constants, or assumption regimes) that would decisively favor their method in this purely theoretical submission. Another review echoed concerns about the technical novelty of the confidence-set construction, asked for optimality evidence (e.g., lower bounds or sharp placements relative to known regret frontiers under correlation), and noted the absence of numerical illustrations that could clarify the price of dependence and the practical value of the proposed analysis. Another reviewer raised pragmatic questions about bias in the noise process and hyperparameter tuning that may depend on mixing information; the rebuttal addressed these at a high level to the reviewer’s satisfaction, though concrete tuning guidance and assumptions remain diffuse in the main text. Overall, the referee panel remains split, with widely varying scores and confidence levels, and the most positive review carries the lowest confidence.

As an afterthought, the AC also notes an additional concern, at a philosophical/modeling level, from a close read. If persistent bias is allowed (as it is in the framework of this paper) -- e.g., $r_t = x_{t,a_t}^\top \theta_\star + \varepsilon_t$ with $E[\varepsilon_t \mid F_{t-1}] = b_t \equiv a>0$ -- the paper must clearly specify the learning objective and the regret notion, since these may differ from the classical mean-zero setting. In particular, if $b_t$ is action- or context-dependent, the observed maximizer $\tilde a_t^\star \in \arg\max_a\{x_{t,a}^\top\theta_\star + b_t(a)\}$ can differ from the latent $a_t^\star$, so one must state whether regret is measured against $x_{t,a}^\top\theta_\star$ or the bias-perturbed mean.

Given these considerations—and noting that reviewer opinions remain split with widely varying scores and confidence (with the most positive review carrying the lowest confidence)—I recommend that the paper not be accepted at the current stage. For a strong revision, I encourage the authors to (i) precisely delineate the dependence classes covered by Assumption 1 (e.g., Markov, mixing) and state whether bias is allowed together with the target regret notion under bias; (ii) work on a clear comparison to the classical martingale/self-normalized framework and a quantitative analysis of spaced subsampling versus the proposed method (conditions under which each succeeds/fails, rates/constants, or a lower-bound separation if available); and (iii) include minimal but telling experiments against baseline strategies to illustrate the theoretical claims. These steps would make the contribution and its scope much clearer to the community.